# CRISPR/Cas9-Mediated Cytosine Base Editing Using an Improved Transformation Procedure in Melon (*Cucumis melo* L.)

**DOI:** 10.3390/ijms241311189

**Published:** 2023-07-07

**Authors:** Hadi Shirazi Parsa, Mohammad Sadegh Sabet, Ahmad Moieni, Abdolali Shojaeiyan, Catherine Dogimont, Adnane Boualem, Abdelhafid Bendahmane

**Affiliations:** 1Université Paris-Saclay, CNRS, INRAE, Université Evry, Institute of Plant Sciences Paris-Saclay (IPS2), 91190 Gif sur Yvette, France; hadi.kharestani@universite-paris-saclay.fr; 2Université Paris Cité, CNRS, INRAE, Institute of Plant Sciences Paris-Saclay (IPS2), 91190 Gif sur Yvette, France; 3Department of Plant Genetics and Breeding, Faculty of Agriculture, Tarbiat Modares University, Tehran 14115-336, Iran; ms.sabet@modares.ac.ir (M.S.S.); moieni_a@modares.ac.ir (A.M.); 4Department of Horticulture, Faculty of Agriculture, Tarbiat Modares University, Tehran 14115-336, Iran; shojaeiyan@modares.ac.ir; 5INRAE, Génétique et Amélioration des Fruits et Légumes (GAFL), 84143 Montfavet, France; catherine.dogimont@inrae.fr

**Keywords:** genetic transformation, gene editing, *CmeIF4E* gene, ethylene

## Abstract

Melon is a recalcitrant plant for stable genetic transformation. Various protocols have been tried to improve melon transformation efficiency; however, it remains significantly low compared to other plants such as tomato. In this study, the primary focus was on the optimization of key parameters during the inoculation and co-culture steps of the genetic transformation protocol. Our results showed that immersing the explants in the inoculation medium for 20 min significantly enhanced transformation efficiency. During the co-culture step, the use of filer paper, 10 mM 2-(N-morpholino)-ethanesulfonic acid (MES), and a temperature of 24 °C significantly enhanced the melon transformation efficiency. Furthermore, the impact of different ethylene inhibitors and absorbers on the transformation efficiency of various melon varieties was explored. Our findings revealed that the use of these compounds led to a significant improvement in the transformation efficiency of the tested melon varieties. Subsequently, using our improved protocol and reporter-gene construct, diploid transgenic melons successfully generated. The efficiency of plant genetic transformation ranged from 3.73 to 4.83%. Expanding the scope of our investigation, the optimized protocol was applied to generate stable gene-edited melon lines using the Clustered regularly interspaced short palindromic repeats/CRISPR-associated protein 9 (CRISPR/Cas9)-mediated cytosine base editor and obtained melon lines with editions (C-to-T and C-to-G) in the eukaryotic translation initiation factor 4E, *CmeIF4E* gene. In conclusion, the optimized melon transformation protocol, along with the utilization of the CRISPR/Cas9-mediated cytosine base editor, provides a reliable framework for functional gene engineering in melon. These advancements hold significant promise for furthering genetic research and facilitating crop improvement in this economically important plant species.

## 1. Introduction

The Cucurbitaceae family comprises many high-value vegetable and fruit crops such as cucumbers, melons, watermelon, zucchini, pumpkin, and squash. Melon (*Cucumis melo* L.) is a diploid species (2*n* = 2*x* = 24 chromosomes) with a relatively small genome and an estimated size of 450 Mbp. This species is known for its high phenotypic variability, particularly in terms of fruit characteristics [1,2,3,4]. Based on Food and Agriculture Organization (FAO) statistics, global melon production and yield have increased by 13.26% and 10.32%, respectively, from 2010 to 2021, while the harvest area of melon has declined by 1.5% [5,6]. With the world’s population growing, ensuring food security necessitates reducing the harvest area and increasing yield [7]. This requires the development of new suitable melon varieties that are resistant to biotic and abiotic stresses.

Conventional breeding methods have been used to develop novel melon varieties with desired traits such as resistance to biotic and abiotic stresses, fruit shape and post-harvest conservation [8,9,10]. These traditional breeding approaches have been extremely successful and remain the basis of plant breeding. However, the efficiency of conventional breeding depends on the existing genetic diversity, which is limited in crops having experienced genetic bottlenecks during domestication. Moreover, conventional breeding often encounters challenges associated with the linkage of undesirable to desirable genetic traits. Thus, these limiting factors make conventional breeding, a long process that can take, on average, a decade to develop a new plant variety. To overcome these limitations, researchers have developed new breeding techniques and biotechnological tools aimed at accelerating the breeding process, accessing new genetic diversity and transferring novel traits. Most of these approaches rely on introducing foreign DNA into plants through *Agrobacterium tumefaciens*-mediated transformation [11,12,13]. Therefore, establishing an efficient plant genetic transformation protocol based on *A. tumefaciens* is critical for melon genetic improvement.

Stable *Agrobacterium*-mediated transformation and plant regeneration are two significant prerequisites for successful plant transformation. Several reports have shown that transformation efficiency in melon is notably low [14,15]. This limitation hinders the application of transgenesis and genome editing for functional genomics studies [16,17]. Different factors affect the *Agrobacterium*-mediated transformation efficiency, particularly those related to the inoculation and co-cultivation medium conditions. In the inoculation medium, high and low *Agrobacterium* inoculum levels, as well as excessively short or long durations of co-cultivation between *Agrobacterium* and the plant explant, have adverse effects on the transformation efficiency [18,19]. Moreover, the strain of *Agrobacterium* and the inoculation procedures, such as immersion of explants into the inoculum or vacuum infiltration are other parameters that affect the plant transformation efficiency [20,21]. The most critical step in *Agrobacterium*-mediated transformation is co-cultivation. Several factors, including thiols, acetophenone and explant type have been tested at this step to improve transformation efficiency in different crops [18,22,23,24]. Optimizing the co-culture duration and temperature enhances the interaction between *Agrobacterium* and plant cells, as well as promotes *vir* gene expression in the co-culture medium [25,26]. Ethylene is a crucial regulator and inhibits *Agrobacterium*-mediated genetic plant transformation. Generally, reducing explant ethylene production using chemical inhibitors such as aminoethoxy-vinylglycin (AVG), silver nitrate (AgNO_3_), and potassium permanganate (KMnO_4_) increases the efficiency of *Agrobacterium*-mediated gene transfer [27,28,29,30,31,32].

Another bottleneck in the transformation and regeneration of Cucurbitaceae species, such as melon and cucumber, is the ploidy level. The ploidy levels of regenerated plants are influenced by the type and age of the starting explants [12,33]. Although both organogenesis and somatic embryogenesis have been used to transform and regenerate melons, direct organogenesis results in fewer spontaneous mutations compared to somatic embryogenesis, making it more suitable for the creation of transgenic plants [21]. Melon organogenesis and embryogenesis are typically performed using leaf and cotyledon explants. However, several studies have utilized cotyledonary explants for direct shoot organogenesis [15,16,17,18,19,20,21,22,23,24,25,26,27,28,29,30,31,32,33,34]. Regeneration of melon plants from cotyledonary explants is more efficient than leaf explants, but it leads to a higher polyploidy level in the regenerated plants [20]. Therefore, it is crucial to consider regeneration efficiency and potential consequences, such as the polyploidy level.

Clustered regularly interspaced short palindromic repeats/CRISPR-associated protein 9 (CRISPR/Cas9) is one of the most powerful genome editing tools for creating double-strand breaks (DSBs) in a target sequence. The plant repairs these DSBs using nonhomologous end-joining (NHEJ) or homology-directed repair (HDR) pathways. NHEJ, an error prone repair pathway, introduces mutations such as insertions or deletions (Indels) at the repair junctions, whereas HDR enables precise repair of DSBs [35]. The CRISPR/Cas9 system has been successfully applied in various model plants and crops to engineer plants with desired traits, primarily leveraging the NHEJ pathway [36,37]. Another approach for plant genome editing, utilizing the HDR pathway, aims to introduce precise and specific mutations at the target site, but the efficiency of HDR is often low [38]. Inducing precise point mutations using the HDR process requires a template DNA carrying the desired sequence. However, DNA delivery within plant cells and nuclei presents challenges and may not be effective in all species. To overcome these limitations, base-editing (BE) approaches have been developed. BE methods, which do not induce DSBs, involve the fusion of a catalytically dead/nikase Cas9 variant (d/nCas9) and DNA-modifying enzymes such as deaminases and demethylases. This serves as an alternative to the low efficiency of the HDR pathway [39,40]. Target-activation-induced cytidine deaminase (Target-AID) is an example of a base-editing construct that consists of the nikase Cas9 (nCas9) linked to *Petromyzon marinus* cytidine deaminase (PmCDA1). It has been used to introduce point mutations into target genes in plants [41,42,43].

Virus resistance is a desirable trait in crops, and in melon, resistance to Potyviridae viruses has been linked to mutations in the eukaryotic translation initiation factor 4E (eIF4E) [11,44,45]. The interaction between the viral protein genome (VPg) of potyviruses and the plant eIF4E is crucial for completing the virus infection cycle [46]. In this study, we report the development of an efficient *Agrobacterium*-mediated melon transformation protocol and the use of a CRISPR/Cas9 base editing approach to generate plants with an edited *eIF4E* gene.

## 2. Results

### 2.1. The Effects of Inoculation Time, Agrobacterium Concentration and Vacuum Infiltration on Explants Transformation Efficiency

Melon explants were transformed using the EHA105 strain of *Agrobacterium tumefaciens*. Each seed was cut into eight sections and inoculated with an inoculation medium containing *A. tumefaciens* expressing the *DsRed* reporter gene. The number of explants expressing DsRed was counted (Figure 1A–D). The results showed that when the inoculation time was reduced to 20 min, the transformation frequency of the *Agrobacterium*-inoculated explants was higher. Compared to a 60 min immersion, a 40 min immersion resulted in higher transformation efficiency, but the difference was not significant (Figure 1G). No significant differences were detected in transformation efficiency between the different concentrations of *Agrobacterium* (Figure 1H), but with the increase in *Agrobacterium* concentration, DsRed expression increased and overgrowth of *Agrobacterium* was observed. The efficiency of transformation was compared between immersion of explants in the inoculum and the vacuum infiltration method (Figure 1E,F) and showed no significant differences between the two methods. Furthermore, significant differences were observed between Khatooni and Charentais mono in both vacuum infiltration and explant immersion in the inoculum (Figure 1I). Based on these results and for further experiments, melon explants were immerged for 20 min in an inoculation medium inoculated with *Agrobacterium* at an OD_600_ of 0.4.

### 2.2. The Filter Paper, Reduce the Temperature and pH Stability in the Co-Culture Medium Increased Transformation Efficiency

To assess the effects of filter paper and agar on solidifying the co-culture medium, we co-cultured Charentais mono and Khatooni cultivars with *Agrobaterium* and showed that the transformation efficiency on filter paper moistened with 15 mL of liquid co-culture medium was higher than that in solidified medium with 8 g/L of agar for both Charentais mono (37.48%) and Khatooni (15.54%) (Figure 2A). The pH stability was another factor in the co-culture medium that affected the transformation efficiency. In both Charentais mono and Khatooni cultivars, the percentage of explants with positive DsRed expression significantly differed between MES+ and MES− treatments (Figure 2B). The stability of pH with 10 mM MES increased the transformation efficiency compared to the co-culture medium without MES. The effect of temperature on the transient expression of DsRed showed that at a temperature of 24 °C, compared to 27 °C, the transformation efficiency was higher for Khatooni (11.9%) and Charentais mono (25%) cultivars (Figure 2C).

### 2.3. Ethylene Inhibitors and Absorbers

#### 2.3.1. Effect of AgNO_3_ on Agrobacterium Growth

AgNO_3_ is an ethylene perception inhibitor, but at high concentrations, it also inhibits the growth of *Agrobacterium* due to its bactericidal properties. To examine the effect of AgNO_3_ on the growth of *A. tumefaciens*, different concentrations of AgNO_3_ were tested. The results demonstrated that at concentrations of 20 and 40 µM AgNO_3_, the growth of *Agrobacterium* was similar to the control (0 µM AgNO_3_). In contrast, at 60 µM AgNO_3_, the growth of *Agrobacterium* was inhibited and stopped five days after incubation. No growth of *Agrobacterium* was observed at 80 and 100 µM AgNO_3_, during the 5 days of incubation period (Figure 2D). Therefore, the concentration of 40 µM AgNO_3_ was selected as the optimal ethylene inhibitor in the co-culture medium.

#### 2.3.2. Effects of Ethylene Inhibitors and Absorber on Transformation Efficiency

The effects of ethylene biosynthesis or perception inhibitors and absorbers on the transient transformation of melon explants were investigated and summarized in Figure 2E. In both Khatooni and Charentais mono melon varieties, all treatments showed a significant difference compared to the control. In Khatooni, AgNO_3_, AVG + AgNO_3_, and KMnO_4_ treatments did not have significant differences, but they resulted in a significant increase in transformation efficiency compared to the control. The addition of 10 µM AVG led to a lower increase in transformation efficiency compared to the AgNO_3_, AVG + AgNO_3_, and KMnO_4_ treatments. In Charentais mono, there was no significant difference between the AVG + AgNO_3_ and KMnO_4_ treatments in terms of transformation efficiency, but both treatments showed significant differences compared to the control. In Charentais mono, both the AVG and AgNO_3_ treatments increased transformation efficiency compared to the control, but the co-cultivation with AVG + AgNO_3_ resulted in a higher increase in transformation efficiency compared to using AVG or AgNO_3_ treatments individually. The highest difference was observed between the control and the KMnO_4_ treatment, with transformation efficiency increasing by 63.64% and 64.29% in Khatooni and Charentais mono, respectively (Figure 2E). Although KMnO_4_, as an absorber of ethylene, had the greatest impact on transformation efficiency, its management in a large-scale tissue culture medium was challenging. So, with respect to the fact that there is no significant difference between KMnO_4_ and AVG + AgNO_3_ treatments, we have chosen AVG + AgNO_3_ for the stable transformation of melon explants.

### 2.4. Optimization of Kanamycin Concentration for Efficient Selection of Transgenic Plant

To optimize the kanamycin concentration for efficient selection of transgenic plants, various concentrations ranging from 0 to 250 mg/L kanamycin were tested. At concentrations of 0, 50, 100, 150, and 200 mg/L, some explants exhibited green coloration, indicating their viability. However, at 250 mg/L, all melon explants turned white and necrosed (Figure 3A). Among the tested concentrations, explants at 0, 50, and 100 mg/L showed observable buds’ formation in the regeneration medium. Conversely, at concentrations of 150, 200, and 250 mg/L, explants were unable to regenerate buds (Figure 3B,C). Based on these results, a kanamycin concentration of 200 mg/L was selected for the stable transformation of melon explants.

### 2.5. Stable Transformation Optimization

Optimal results of transient transformation were used for the stable transformation of melon explants. The Charentais mono and Khatooni explants were immersed in *Agrobacterium* inoculum for 20 min with an OD_600_ 0.4 and then transferred to co-culture medium. After 3 days, explants were washed and transferred to the selective medium. The regenerated shoots from melon explants were selected after 6–8 weeks in the selective medium and transferred to the rooting medium with 150 mg/L kanamycin and 200 mg/L timentin for 4–6 weeks. To reduce the chimerical plants, 150 mg/L kanamycin was used in the rooting medium (Figure 4).

Finally, 62 plants regenerated from Charentais mono explants, while 79 plants regenerated from Khatooni explants. Among these, five Charentais mono plants and eight Khatooni plants expressed DsRed, as shown in Table 1. All DsRed-positive shoots successfully developed roots after being transferred to the rooting medium. DsRed expression intensity varied significantly among melon organs. The highest DsRed expression was observed in male flowers (Figure 4I). Moreover, DsRed expression was observed in the fruit and seed of some Charentais mono transgenic lines (Figure 4K,M). The results of flow cytometry analysis showed that in Charentais mono 100% and in Khatooni 87.5% of the rooted plants were diploid (Figure 5, Table 1).

Since transgenic lines of the Khatooni cultivar were unable to produce female flowers in the greenhouse, only Charentais mono diploid plants were self-pollinated after acclimation, resulting in the first generation of diploid transgenic plants. The transformed/non-transformed plant ratio was assessed using the chi-square test in the progeny of some Charentais mono transgenic lines to evaluate the number of T-DNA insertions (Table 2). The chi-square test showed that the deviation from the expected 3:1 segregation ratio was not statistically significant, suggesting the insertion of only one copy of the T-DNA into the melon genome.

### 2.6. Genome Editing Using CRISPR-Cas9 Cytidine Deaminase Fusion

CRISPR/Cas9-mediated base editing (BE) for genetic engineering has been used extensively in different crops. Because melon is recalcitrant to genetic transformation, no experiment reports the application of CRISPR/Cas9-mediated base editing to melons. In our study, we tested whether Target-AID could induce single mutations in melon genome, and if our optimized protocol is efficient for melon genetic transformation with CRISPR/Cas9-mediated base editing constructs. Two Target-AID-DsRed constructs, including sgRNA targeting the first and second exons of the *eIF4E* gene were used for transformation of Charentais mono, Vedrantais, and Samsuri melon varieties (Figure 6A, Table 3). To detect the transgenic plants on the selective medium in the first steps of transformation, the DsRed reporter gene was inserted into the Target-AID construct (Target-AID-DsRed) (Figure 6B). The sgRNAs were inserted into the Target-AID-DsRed vector under the control of the *Arabidopsis* U6 promoter and the Cas9 coding sequence under the control of the parsley ubiquitin promoter (PcUbi) (Table 3). The resulting construct was transferred into melon explants by the optimized *Agrobacterium*-mediated transformation method.

Regenerated shoots were chosen based on resistance to kanamycin and DsRed expression. The transformation efficiency calculated based on rooted plants in the greenhouse ranged from 3 to 8%. The highest and lowest transformation efficiencies were observed in Samsuri and Vedrantais, respectively. The number of transgenic plants obtained with sgRNA-2 was higher than that obtained with sgRNA-1 (Table 4). To identify the mutations induced by the CRISPR/Cas9 system in the T_0_ generation, the target sequences were sequenced (Figure 6C).

The results of CRISPR-ID analysis revealed that the genomes of nine T_0_ plants were edited as chimeric or heterozygotic. To assess the efficiency of base-editing in the target sequences in the T_0_ generation, PCR products were cloned into the pJet1.2 vector. For each line, 5–10 colonies were picked up and sequenced. Most lines showed high efficiency in cytidine to thymine (C to T) substitution. Interestingly, no insertion was observed in the T_0_ lines at the target sequences. Deletion size ranged from 5 to 9 bp in the transgenic lines obtained with sgRNA-1, while no Indel was observed in the edited plants with sgRNA-2. Sequencing data showed that the genome editing frequency ranged from 20% to 40% in the T_0_ generation of the three genome-edited melon varieties (Table 5). The base editing window observed was between −17 and −10 from the PAM sequence (Figure 6C). To evaluate the mutation heritability induced by CRISPR/Cas9, two lines, #3-3 and #3-6, of the genome-edited Samsuri variety transformed by the sgRNA-2 construct were self-pollinated. Target sequences were amplified with specific primers in at least five plants per line, and the T_1_ progeny was analyzed. In the T_1_ generation, as expected, homozygous mutagenesis was observed. In the first line, the *CmeIF4E^C322T/C323T^* substitutions resulted in the change of the CCA to TTA codon (P108L), while *CmeIF4E^C322T/C323G^* substitutions in the second line led to the stop codon (CCA to TGA) (Figure 6D). CRISPR/Cas9-mediated base editing can be used to change a single nucleotide at the genome level without affecting the growth of the plant. The use of base-editing techniques in melons is highly dependent on an efficient transformation protocol. Our results show that the optimized protocol is efficient for transferring the CRISPR/Cas9 construct to the melon genome. Base editing is also a rapid and precise method for changing a single nucleotide in the genome. The results of this study demonstrated that base editing is heritable in subsequent generations and could be used to accelerate crop improvement.

## 3. Discussion

Different factors in the inoculation and co-culture media were evaluated, and optimal conditions were used for the stable transformation of different melon varieties with pMOG800 and Target-AID-DsRed constructs. In our study, the explants were immersed for 20 min in the *Agrobacterium* with an OD_600_ of 0.4 for stable transformation. Zhang and Finer [26] reported that longer immersion times of explants and high concentrations of *A. tumefaciens* could induce the plant defense system and reduce the transformation efficiency. Rajagopalan and Perl-Treves [47] found that longer inoculation times increased GUS expression in the transformed explant but decreased plant regeneration and survival rate. These reports support our observations.

The vacuum infiltration was used to improve the penetration of *Agrobacterium* into the epidermal cells, which are the source of organogenesis in melon [20,48]. Vacuum infiltration has been used to enhance the efficiency of transformation in various plants such as rice [49], citrus [50], cowpea [48], cucumber [21], and melon [51]. By creating negative pressure, vacuum infiltration facilitates the removal of gases from the explant interior, allowing the *Agrobacterium* suspension to effectively infect the target tissue when the vacuum is released, and the pressure rapidly increases [52]. In our study, the use of vacuum infiltration for *Agrobacterium* inoculum did not significantly improve transformation efficiency. The pressure and duration of the vacuum play a role in enhancing the efficiency of transformation, but it depends on the type of plant and explant. Hooghvorst et al. [51] successfully improved melon genetic transformation using vacuum infiltration on two-day-old melon cotyledons. In our study, we used a one-day-old cotyledon explant, which may require higher pressure and a longer duration of vacuum to enhance transformation efficiency [52].

Ethylene is an important plant hormone that plays a critical role in many developmental processes, including fruit ripening, abscission, senescence, growth, flowering, sex expression, and plant-microbe interactions [53,54]. Ethylene acts as a suppressor of *vir* genes in *Agrobacterium* and negatively affects melon transformation. Ezura et al. reported that the use of 10 µM AVG in the co-culture medium reduces ethylene production and enhances *Agrobacterium*-mediated gene transfer [27]. Another study has shown that using KMnO_4_ and AgNO_3_ in the co-culture medium of *Leucojum aestivum* L. reduces ethylene production and increases the dry weight of the callus [29]. In *Prunus avium*, AVG, and AgNO_3_ had a strong effect on regeneration [30]. Malambane et al. [32] reported that AgNO_3_ and AVG reduced ethylene accumulation in wild watermelon cotyledon explants inoculated with *Agrobacterium* strain EHA105. In our study, the addition of ethylene inhibitors and absorbers, such as AVG, AgNO_3,_ and KMnO_4_, in the co-culture medium, reduced ethylene production and improved the efficiency of transformation. The highest transformation efficiency was observed using KMnO_4_ or a combination of AgNO_3_ and AVG. However, due to technical reasons, 40 µM AgNO_3_ and 10 µM AVG were added to the co-culture medium for stable melon transformation.

Kanamycin has been widely used for selecting transgenic melon plants in many protocols. The appropriate concentration of kanamycin for selection depends on genotypes and cultivars, and optimization is necessary before transformation [15,24,55,56]. Choi et al. [15] reported that melon explants retained their photosynthetic activity and their green color when exposed to kanamycin concentrations ranging from 50 to 750 mg/L. Ren et al. [13] found that at kanamycin concentration of 100–150 mg/L, callus tissue and shoot primordia were produced, but at 150–200 mg/L kanamycin, plant shoot regeneration was completely inhibited. In our study, a kanamycin concentration of 200 mg/L was used in the selective culture medium, effectively blocking shoot regeneration in non-transformed control explants. However, during the genetic transformation experiments, we observed the false-positive (escaped) plants. These observations agree with previous reports that the optimal kanamycin concentration in the regeneration medium could not prevent the growth of false-positive plants in the selective medium after transformation [24]. The rate of escaped plants during melon transformation was high and affected by several factors. One reason is the transient expression of the resistant gene in the cells [57]. Moderate resistance of melon cotyledons to kanamycin is another factor contributing to non-transformed escape in the selective medium [58]. Using a high concentration of kanamycin in the rooting medium could help to reduce the number of chimeric and false-positive plants, since most chimeric shoots do not survive on the kanamycin selection medium and die before or during rooting. Another approach to reducing the number of false-positive plants is the use of reporter genes. Reporter genes, such as DsRed, provide the advantage of selecting transformed buds/plants and eliminating non-transgenic plants at an early stage of transformation, saving energy, time, and labor, while improving transformation efficiency in melon. The combination of antibiotics and reporter genes enables uniform selection of transformed plants.

The choice of a suitable explant is the first step in developing a new plant transformation and regeneration protocol. Different explants such as hypocotyl, cotyledon, leaf, quiescent seed, and root have been used for somatic embryogenesis and shoot induction in melon [14,20,59,60,61,62,63,64]. Cotyledons have a greater capacity for developing buds into shoots compared to leaf explants, but they exhibit a higher frequency of polyploid cells [14,20,65]. Guis et al. [14] reported that 18.6% and 84.6% of regenerated melon plants were diploid when using two-day-old cotyledons and ten-day-old leaves, respectively. In our study, the nuclear DNA content of transgenic plants was 2C, 4C, and 8C. The 2C and 4C correspond to the G1 and G2 nuclear phases of the cell cycle, respectively. The same results were obtained in non-transformed control plants grown from seeds in the greenhouse. These results showed that during seedling development, the DNA content of cotyledonary cells changes [33]. Indeed, endoreduplication occurs not only in in vitro but also in the greenhouse and field, leading to polyploidy [66]. Many *Agrobacterium*-mediated transformation protocols have used two–seven-day-old cotyledons as explants for embryogenesis and indirect organogenesis via callus. However, these explants have disadvantages such as high polyploidy, somaclonal variation, and time-consuming plant generation from callus, all of which affect transformation efficiency. In our study, the quiescent seed of melon was used for transformation and flow cytometry results showed that most of the transformed plants were diploid. Moreover, direct shoot regeneration from quiescence seed without a callus phase is desirable for transformation using the *Agrobacterium*-mediated transformation method for several reasons: (1) no somaclonal variation, (2) low polyploidy, (3) early selection of uniform plants, and (4) reduced labor, cost, and energy requirements.

Plant resistance to viruses such as potyviruses, presents a challenge in conventional breeding due to limited natural resistance resources [67]. Potyviruses encode a small set of proteins responsible for hijacking plant host resources for their replication. The viral protein genome-linked (VPg) has the ability to recruit plant translation initiation factors eIF4E, eIFiso4E, or both, to capture the host translational machinery and complete the virus infection cycle [68,69,70,71]. Knocking out *eIF4E/iso4E* genes is a promising strategy for developing potyvirus-resistant plants. However, knockout of factors such as eIF4E/iso4E that play essential roles in cell biological processes can have induce adverse effects on plant development [11,72]. CRISPR/Cas9-mediated cytidine and adenosine base editing (CBE or ABE) can induce modifications in protein amino acids that are not lethal for plant but could affect the interaction of VPg and eIF4E. CBE and ABE methods offer an alternative to conventional approaches by mimicking *eIF4E* resistance alleles, such as *nsv* [44]. In our study, we successfully substituted Cs with Ts or Gs using a CBE approach, generating missense mutations in targeted amino acids. This result is consistent with a previous report in tomato, where Target-AID results in C to T and G to A substitutions [42].

## 4. Materials and Methods

### 4.1. Preparation of Explants

In this study, two cultivars of *Cucumis melo* var. Khatooni and Charentais mono were utilized for both transient transformation and stable transformation. Additionally, the Samsuri, Charentais mono and Vedrantais cultivars were employed to investigate transformation efficiency using the Target-AID-DsRed construct. For the preparation of explants, the seed coats were carefully removed using a scalpel. Subsequently, the peeled seeds’ surfaces were disinfected for one minute with 70% ethanol and rinsed with sterile distilled water. Sterilization of the seeds was achieved by treating them with 3% (*w*/*v*) sodium hypochlorite solution for 30 min, supplemented with one drop of Tween 20, followed by five rinses with sterile distilled water. Afterwards, the sterilized seeds were placed in a 10 cm plastic Petri dish containing 15 mL of water and three layers of Whatman No. 1 filter paper (Merck, Darmstadt, Germany) and incubated overnight in the dark at 28 °C. Finally, using tweezers, the radicle section of each seed was carefully removed, and the remaining portion of each seed was sliced transversally and longitudinally to obtain eight equally sized explants.

### 4.2. Agrobacterium Strain, Binary Vector and Inoculation Medium

The *A. tumefaciens* strain EHA105 and the binary vector pMOG800 harboring the DsRed reporter gene (kindly provided by Prof. Burgos [24]) were used for the transformation of melon. The kanamycin resistance (*npt*II) gene, under the control of the NOS promoter, was used to select the transformed plants in the selection medium. For the *Agrobacterium* culture, a single colony of *Agrobacterium* was inoculated into 15 mL of Luria–Bertani (LB) medium (pH 7.0) containing 50 mg/L kanamycin and 30 mg/L rifampicin. The culture was incubated at 28 °C on a rotary shaker at 140 rpm. Once the optical density at 600 nm (OD_600_) reached 0.4–0.5, the *Agrobacterium* suspension was centrifuged and diluted to an OD_600_ of 0.4 using an inoculation medium including 2.2 g/L MS salt medium plus B5 vitamins, 1% (*w*/*v*) sucrose. To efficiently induce the *vir* genes, the diluted *Agrobacterium* inoculum was gently shaken at 28 °C for two hours [73].

### 4.3. Levels, Durations, and Methods of Explants Inoculation with Agrobacterium

For the inoculation step, three experiments were designed to evaluate different factors. In the first experiment, three levels of inoculation (OD_600_ 0.4, 0.6, and 0.8) were tested. In the second experiment, three durations of inoculation (20, 40, and 60 min) were examined. Lastly, the third experiment involved evaluating two methods for inoculating the explants. In the first method of incubation, the explants were immersed in the *Agrobacterium* inoculum for 20 min. The second method, based on Bowen et al. [74] with some modifications, involved placing the explants on the shaker for five minutes and then transferred to a 20 mL syringe containing 10 mL of inoculum medium with an OD_600_ of 0.4. The syringe’s distal end was sealed with a cap after removing excess air. Vacuum infiltration was carried out by slowly pulling the plunger from 10 to 20 mL, followed by gentle shaking of the syringe for five minutes. The plunger was then slowly returned to 10 mL. It was repeated twice. After inoculation, explants were blotted dry on sterilized filter paper and transferred to the co-culture medium.

### 4.4. Co-Culture Medium Experiments

Four experiments were carried out in the co-culture medium to examine the effects of various factors: solidifying factor (0.8% agar vs. three layers of sterilized Whatman No. 1 filter paper moistened with 15 mL of co-culture medium), MES (addition of 10 mM MES to the co-culture medium), temperature (24 °C and 27 °C), and ethylene inhibitor and absorber. The optimal concentration of AgNO_3_ for *A. tumefaciens* growth in the co-culture medium was also determined. In order to determine the optimal concentration of AgNO_3_, a single colony of *A. tumefaciens* strain EHA105 was grown in LB liquid medium until reaching an OD_600_ of 0.4–0.5. It was then centrifuged at 4000× *g* and 22 °C and diluted in 20 mL of LB medium to achieve a final OD_600_ of 0.4. Subsequently,10 µL of the medium was cultured in six-well polystyrene (35 mm) flat-bottom microplates containing 7 mL of LB solid medium including different concentrations of AgNO_3_ (0, 20, 40, 60, 80, 100 µM). The microplates were incubated in the dark at 28 °C for four days, and the growth rates of bacterial colonies were estimated based on visual observation. The effect of Aminoethoxyvinylglycine (AVG) and silver nitrate (AgNO_3_) as inhibitors and KMnO_4_ (2 g in an open tube within the Petri dish) as an absorber of ethylene in the co-culture medium was examined on Charentais mono and Khatooni cultivars [27,29,75]. After incubation of melon explants by *Agrobacterium*, the explants were blotted dry on Whatman No. 1 filter paper and placed in the dark at 24 °C for 72 h. The co-culture medium used included 2.2 g/L MS salt medium plus B5 vitamins, 2% (*w*/*v*) sucrose, 0.8% (*w*/*v*) agar, 4.4 µM BAP and 3 µM MES.

### 4.5. Effect of Kanamycin on Explant Growth

The optimal concentration of kanamycin is crucial for minimizing the occurrence of escape and chimeric plants. In this study, different concentrations of kanamycin (0, 50, 100, 150, 200, and 250 mg/L) were utilized in the selective medium to select transgenic Charentais mono plants. The explants were cultured in Petri dishes containing 30 mL of regeneration medium and supplemented with different concentrations of kanamycin. The adaxial surface of the explants was in contact with the regeneration medium. After four weeks, the survival rate of explants, and the number of shoots in the selective medium were carefully recorded.

### 4.6. Detection of DsRed Expression by Fluorescence Stereomicroscopy

One week after transferring the explants to the regeneration medium, the transient expression of DsRed was assessed using a Leica MZ16F stereomicroscope (Leica, Wetzlar Hesse, Germany). The microscope was equipped with different filter sets to detect DsRed in the explants. The DsRed module contains a 545/30 nm excitation filter and a 620/60 nm emission filter. To capture the images, a Leica DFC300 FX color camera was employed.

### 4.7. Stable Transformation

Optimal results from transient transformation were utilized for the stable transformation of melon explants. The quiescent seeds of Charentais mono and Khatooni were used for stable transformation. Following seed sterilization, each seed was divided into eight sections. The explants were inoculated with a diluted *Agrobacterium* inoculum with an OD_600_ of 0.4 for 20 min. After blotting them dry, the explants were transferred to Petri dishes containing three layers of filter paper and 15 mL of co-culture medium. The co-culture medium comprised 2.2 g/L MS salt medium plus B5 vitamins, 2% (*w*/*v*) sucrose, 4.4 µM BAP, 40 µM AgNO_3_, 10 µM AVG, and 10 mM MES. After three days, green and swollen explants were selected and washed with water including 200 mg/L timentin. They were then blotted dry on sterilized filter paper. The explants were transferred to the selective medium containing MS salt medium-plus B5 vitamins, 3% (*w*/*v*) sucrose, 0.8% (*w*/*v*) agar, 4.4 µM BAP, 4 µM copper (II) sulfate (CuSO_4_), 3 µM MES, 200 mg/L timentin and 200 mg/L kanamycin. The explants were sub-cultured into fresh media every two weeks. After 4–8 weeks, regenerated plants that expressed DsRed were excised and transferred to rooting media containing 2.2 g/L of MS salt plus B5 vitamin, 10 g/L sucrose, 7 g/L agar, 150 mg/L kanamycin, and 200 mg/L timentin. In order to minimize chimeric plants, 150 mg/L kanamycin was used in the rooting medium. The rooting plants were acclimatized and cultivated in a greenhouse at a day/night temperature of 28/23 °C (day/night temperature), a 16/8 h light/dark cycle, and 50% humidity. All of the media used in this study were autoclaved at 121 °C for 20 min, with the pH adjusted to 5.7–5.8 before autoclaving. Additionally, all hormones and antibiotics were added to the cooling medium by filter sterilization. After 3 days, explants were washed and transferred to selective medium. Regenerated shoots from melon explants were selected after 6–8 weeks on the selective medium and transferred to the rooting medium containing 150 mg/L kanamycin and 200 mg/L timentin for 4–6 weeks.

### 4.8. Flow Cytometry

The flow cytometry was used to compare the relative nuclear DNA content between melons derived from transformation and wild types. The 2C peak from the nuclei of young leaves of diploid plants was used for calibration. Melon samples were prepared from the third leaf (2–3 cm long) below the shoot apex of acclimatized plants grown in the greenhouse. The plant material was placed in a Petri dish containing 500 µL of extraction buffer, which consisted of 45 mM MgCl_2_, 30 mM sodium citrate, 60 mM MOPS at pH 7.2. The material was chopped into 2 mm sections using a razor blade. Additionally, triton at the final concentration of 0.1% (*w*/*v*) was added to the Petri dishes. The nuclei suspension was filtered through Mesh nylon filter (60 µm mesh size) to remove cell debris. Subsequently, 100 µL of a propidium iodide stock solution (1 mg/mL) was added to stain the nuclei mixture. The nuclei mixture was incubated for two minutes at room temperature and then analyzed by a flow cytometry apparatus (Partec, software Flomax v2.4d).

### 4.9. Statistical Analysis

All experiments were designed using a completely randomized block design (CRBD) with two or three replicates. The data were statistically analyzed using analysis of variance (ANOVA) and Duncan’s multiple range test (*p*-value < 0.05) by IBM SPSS Statistics 23.0 (IBM, Armonk, NY, USA). Chi-square analysis was used to compare marker segregation ratios of progenies produced after self-pollination of some transgenic lines to expected values using IBM SPSS Statistics 23.0 (IBM, United States).

### 4.10. Amplifying the DsRed Reporter Gene and Cloning to Target-AID Vector

DsRed reporter gene under the control of CsVMV 35s promoter and NOS terminator was amplified by PCR from the pGGF010 vector and cloned into Target-AID vector. Two primers were designed by the Primer3plus online software v2.5.0 (primer3plus.com), and the restriction sites of *Pac*I-HF and *Bam*HI-HF were added to 5′ of forward and reverse primers, respectively (Table 3). PCR amplifications were performed in a 50 µL reaction mixture, following the cycle program: initial denaturation at 98 °C for 1 min, followed by 35 cycles of 98 °C for 15 s (denaturation), 65 °C (annealing) for 20 s, and 72 °C (extension) for 40 s. The final extension was performed at 72 °C for 5 min. The DsRed fragment and Target-AID vector were digested by the *Pac*I-HF and *Bam*HI-HF restriction enzymes. The synthesized fragment was ligated into the digested binary vector using T4 DNA ligase (Target-AID-DsRed) (New England Biolabs, Ipswich, MA, USA). The reaction mixture was transformed into TOP10 chemically competent *E. coli* and the bacteria were grown overnight at 37 °C on LB plates containing 100 mg/L spectinomycin.

### 4.11. The sgRNA Designing and Cloning in the Target-AID-DsRed Vector and Agrobacterium Transformation

The CRISPR RGEN online software (http://www.rgenome.net/ (accessed on 23 April 2023)) was used to design sgRNA [76]. Based on the lowest off-target and the highest score, two sgRNAs targeting the first and second exons of the *eIF4E* gene were chosen. To clone the sgRNA sequences targeting the *eIF4E* gene of melon into Target-AID-DsRed, overlap PCR reactions were used. Firstly, a 344 bp fragment containing the 3′ end of the AtU6-26 promoter, the 20 bp guide, the sgRNA scaffold and the *Spe*I restriction site were synthesized by Target_F_320 and sgRNA-R primers (fragment I). In the second PCR reaction, the *Asc*I restriction site and 10 bp of sgRNA were synthesized by the AtU6-F and AtU6-R primers (fragment II). In the 3rd PCR, the synthesized fragments I and II were amplified by the AtU6-F and sgRNA-R primers. The amplified fragment and Target-AID-DsRed vector digested using *Asc*I and *Spe*I restriction enzymes. The amplified fragment was ligated into the digested binary vector using T4 DNA ligase (New England Biolabs, Ipswich, MA, USA). The reaction mixtures were transformed into TOP10 chemically competent *E. coli* cells. The bacteria were grown overnight at 37 °C on LB plates containing 100 mg/L spectinomycin. Five moderate size colonies were picked up and cultured in LB liquid medium containing 100 mg/L spectinomycin. Plasmid extraction was performed by the Plasmid DNA Mini Kit according to the description provided by Omega BioTek. The sequences and directions of sgRNAs and the U6 promoter were confirmed by the Sanger sequencing method (Eurofins genomics). Plasmids were transferred into the *Agrobacterium tumefaciens* EHA 105 strain by the heat shock method.

## 5. Conclusions

In conclusion, we have developed an optimized protocol for efficient transformation of various melon varieties using pMOG800 and Target-AID-DsRed constructs. Our findings suggest that specific conditions, such as a 20 min immersion of explants in Agrobacterium with an OD600 of 0.4, the use of filter paper, 10 mM MES, the combination of AVG and AgNO_3_, and a temperature of 24 °C, significantly enhance melon transformation efficiency. Kanamycin at 200 mg/L effectively selects transformed plants, although additional strategies are needed to reduce false-positive occurrences. The use of reporter genes such as DsRed aids in the early selection and elimination of non-transgenic plants. Quiescent melon seeds serve as successful explants, offering advantages such as low polyploidy, early uniform plant selection, and reduced labor and cost. Additionally, we successfully applied CRISPR/Cas9-mediated base editing to introduce *eIF4E* gene mutations, showing promise for developing potyvirus-resistant melon plants. Overall, our optimized transformation protocol, along with CRISPR/Cas9-mediated base editing, provides a valuable tool for expediting melon breeding and enhancing virus resistance.

## Figures and Tables

**Figure 1 ijms-24-11189-f001:**
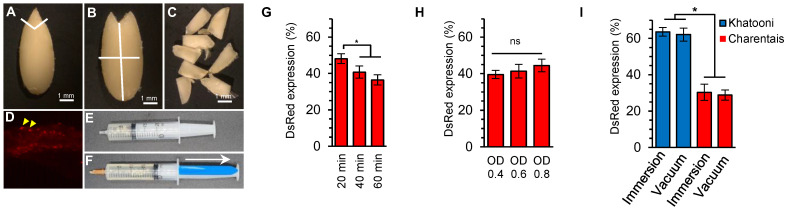
Explant preparation, vacuum infiltration, and the effect of different factors in the inoculation medium on the transformation efficiency of melon cotyledon explants using *Agrobacterium.* (**A**) Melon quiescent seed without seed coat. (**B**) Melon quiescent seed without the radicle. In (**A**,**B**), lines indicate the cutting axis. (**C**) Explant type used for genetic transformation. (**D**) *DsRed* expression in transformed explant. The yellow arrows show the spots of *DsRed* expression. (**E**) Melon explants and *Agrobacterium* suspension in the 20 mL syringe. (**F**) Vacuum infiltration of *Agrobacterium* into melon cotyledon explants. (**G**,**H**) The percentage of explants expressing *DsRed* at different inoculation times (**G**) and concentrations of *Agrobacterium* (**H**). (**I**) Transformation efficiency of melon explants using immersion versus vacuum infiltration methods. Values are the means ± SD from three replicates for each treatment; ns, no statistically significant difference. * *p* ≤ 0.05 (Duncan’s multiple range test).

**Figure 2 ijms-24-11189-f002:**
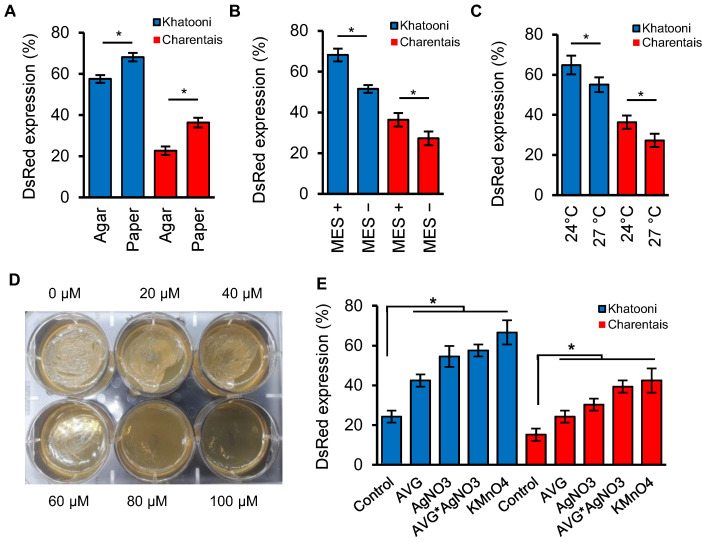
The effect of co-culture components on transformation efficiency and AgNO_3_ on *Agrobacterium* growth. (**A**–**C**) Effect of agar and paper solidifying co-culture medium agents (**A**) of 10 Mm MES (**B**) and temperature (**C**) on explant transformation efficiency. (**D**) *Agrobacterium* growth on LB-agar media supplemented with different concentrations of AgNO_3_. (**E**) The effect of ethylene inhibitor and absorber on explant transformation efficiency in Khatooni and Charentais mono. * *p* ≤ 0.05 level (Duncan’s multiple range test). The values represent means from three replicates for each treatment, bar = ±S.D.

**Figure 3 ijms-24-11189-f003:**
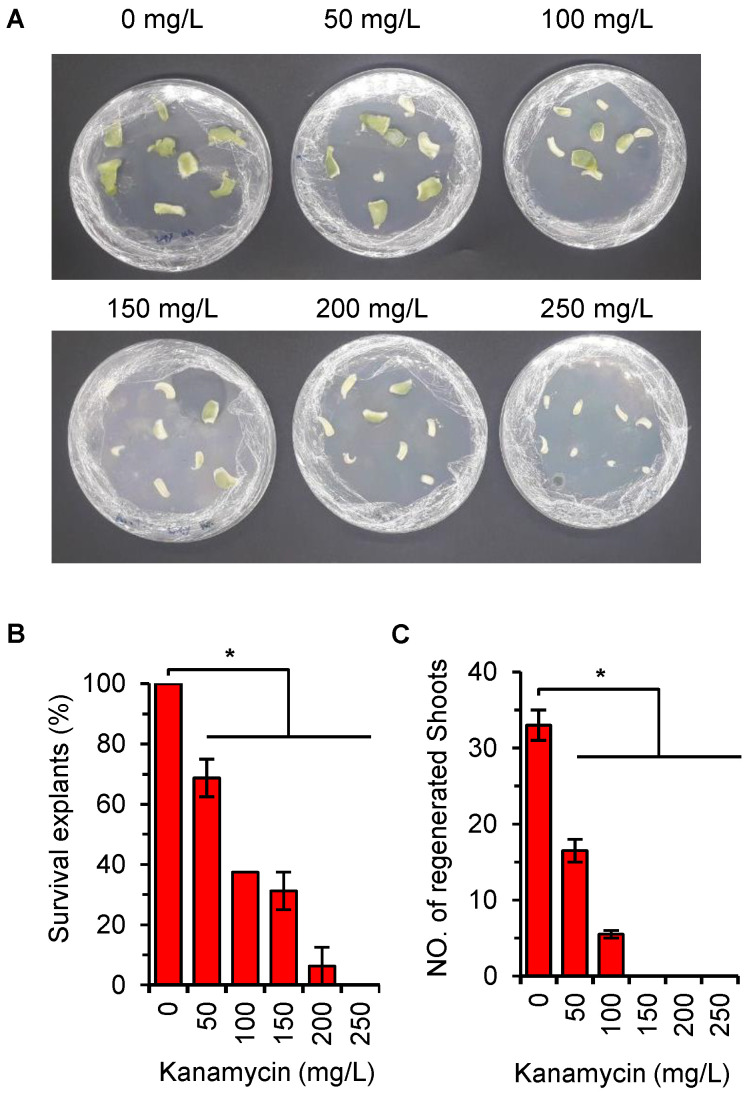
Kanamycin sensitivity assay of Charentais mono explants. (**A**) Melon explants on the selective medium containing different concentration of kanamycin. (**B**) Number of survived explants (%) on the selective medium. (**C**) Shoot regeneration (%) on the selective medium. * *p* ≤ 0.05 level (Duncan’s multiple range test). Values represent means from two replicates for each treatment, bar = ±S.E.

**Figure 4 ijms-24-11189-f004:**
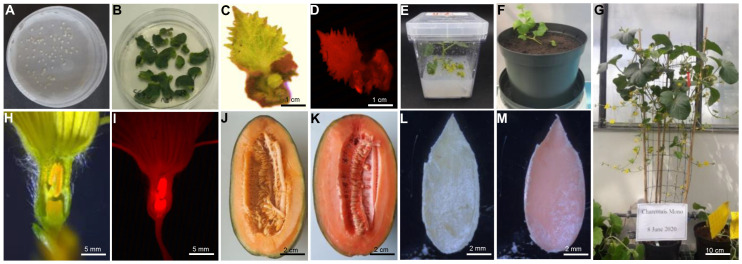
Transformation and regeneration of melon transformed with the pMOG800 vector. (**A**) Explant preparation. (**B**) Placement of explants on the selection medium. (**C**,**D**) Detection of transformed plants by stereomicroscope under white and ultraviolet (UV) light, respectively. (**E**) Transfer of explants to magenta box. (**F**) Acclimatization of plants in a hot chamber. (**G**) Transfer of plants to the greenhouse. (**H**,**I**) Transformed male flower under white and ultraviolet (UV) light. (**J**) Non-transformed fruit under white light. (**K**) Transformed fruit under white light. (**L**) Non-transformed seed under white light. (**M**) Transformed seed under white light.

**Figure 5 ijms-24-11189-f005:**
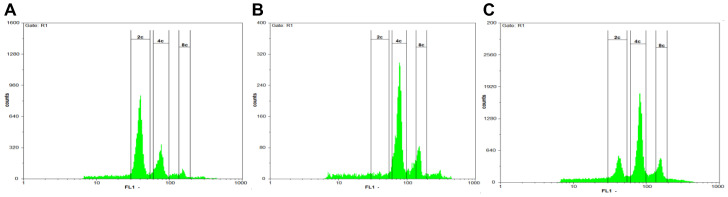
Flow cytometry histograms of transgenic and non-transgenic melons. (**A**) Control plant. (**B**) Diploid regenerated plant. (**C**) Tetraploid regenerated plant. The graph’s letter “c” refers to the number of haploid chromosomes.

**Figure 6 ijms-24-11189-f006:**
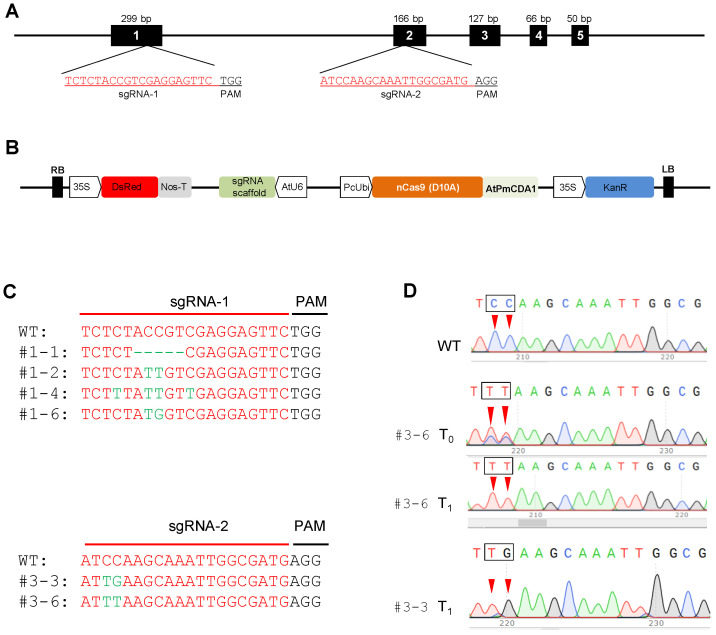
The CRISPR-nCas9-cytidine deaminase edition of *CmeIF4E*. (**A**) Schematic view of the sgRNA target site in the *eIF4E* gene. Black boxes represent the exons. Single guide RNAs, sgRNAs, are indicated in red. PAM sites (5′-NGG-3′) are indicated in black. (**B**) T-DNA physical map of the Target-AID-DsRed binary vector. (**C**) Alignment of mutated plants for sgRNA-1 and sgRNA-2. Black text, PAM; green text, edited nucleotide; #1-1, #1-2, #1-4, #1-6, #3-3, and #3-6 indicate the different edited plants. (**D**) Sanger sequences of WT and edited plants. Edited sites are highlighted by red arrows.

**Table 1 ijms-24-11189-t001:** Transformation efficiency and number of diploid plants produced from Charentais mono and Khatooni cultivars.

Cultivar	Explants	Regeneration (%) ^1^	Escapes ^2^	Rooted Lines	Diploid	T.E ^3^
Charentais mono	134	62 (46.3%)	55	5	5	3.73
Khatooni	145	79 (54.5%)	70	8	7	4.83

^1^ Regeneration: number of explants with at least one regenerated bud. Between brackets the regeneration rate: regeneration/total number of infected explants × 100. ^2^ Plants that grow on selective medium but did not transform. ^3^ Transformation efficiency: the number of rooted lines/total number of explants × 100.

**Table 2 ijms-24-11189-t002:** Segregation analysis of transgene expression in the Charentais mono T_1_ progeny of the transformed lines.

Nb of Progenies	Transformed	Non-Transformed	Segregation Ratio	χ^2^	*p*-Value
235	182	53	3:1	0.75	0.39
186	147	39	3:1	1.61	0.20

**Table 3 ijms-24-11189-t003:** Primers used in this study.

Primer Name	Prime Sequence 5′ to 3′	Amplified Region
For3melon	GGACGGATTGGTTTTAGGGTTC	eIF4E target
eIF4EIIR	CTGTTCTCCGATCATAGCAAGC	eIF4E target
sgRNA-R	CTTTGTACAAAAAAGCAGGCG	sgRNA
eIF-Target-1-R	CTAGAGTCGAAGTAGTGATTG**TCTCTACCGTCGAGGAGTTC**GTTTTAGAGCTAGAAATAGCAAG	sgRNA
eIF-Target-2-R	CTAGAGTCGAAGTAGTGATTG**ATCCAAGCAAATTGGCGATG**GTTTTAGAGCTAGAAATAGCAAG	sgRNA
AtU6_R	GCCATAGAAAAGTTGGGTG	AtU6 promoter
AtU6-F	CAATCACTACTTCGACTCTAG	AtU6 promoter
PacI-For-DsRed	AACAGGGTAATttaattaaAGAAGGTAATTATCCAAGATGTAGCATC	DsRed
BamHI-Rev-DsRed	TTGATCACTAGTggatccTTTATCCTAGTTTGCGCGCTATAT	DsRed

The bold letters correspond to the sgRNA sequences.

**Table 4 ijms-24-11189-t004:** Transformation efficiency and number of plants produced from different melon genotypes and sgRNA.

SgRNA Target	Melon Variety			Total ^2^
	Samsuri	Charentais Mono	Vedrantais	
sgRNA1-272	7 (7%)	5 (5%)	3 (3%)	15 (5%)
sgRNA2-320	8 (8%)	5 (5%)	5 (5%)	18 (6%)
**Total ^1^**	15 (7.5%)	10 (5%)	8 (4%)	33 (5.5%)

^1^ Efficiency of transformation based on variety: No. of transgenic plants/200 × 100. ^2^ Efficiency of transformation based on sgRNA: No. of edited plants/300 × 100.

**Table 5 ijms-24-11189-t005:** Efficiency of base editing in T_0_ transgenic plants.

SgRNA Target	Melon Variety			Total ^2^
	Samsuri	Charentais Mono	Vedrantais	
sgRNA1-272	2 (28.57%)	1 (20%)	1 (33.33%)	4 (26.67%)
sgRNA2-320	2 (25%)	2 (40%)	1 (20%)	5 (27.78%)
**Total ^1^**	4 (26.67%)	3 (30%)	2 (25%)	9 (27.3%)

^1^ Efficiency of transformation based on variety: No. of edited plants/transgenic plants × 100. ^2^ Efficiency of transformation based on sgRNA: No. of edited plants/transgenic plants × 100.

## Data Availability

All the data and plant materials in relation to this work can be obtained through contacting the corresponding authors.

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
