# Peer review of "CRISPR/Cas9-Mediated Cytosine Base Editing Using an Improved Transformation Procedure in Melon (Cucumis melo L.)"

_ijms, 2023, doi:10.3390/ijms241311189_

Round 1

Reviewer 1 Report

This is an interesting work, which should be published in IJMS. But there are some major comments, especially in writing statements.

1. The English of this paper needs to be greatly improved. Many sentences are not rigorous and standardized, which made them hard to be understood. Line 26, 119,

2. Only the results should be placed at the part of Results, no description of methods, Line 121-124, 337-339. No description of backgrounds, Line 204-205, 316-318.

3. Line 129-137? The writing formats of the whole paper should be cheeked carefully before submitting by the authors, especially the corresponding author!

4. Figure 1, the title does not match the contents. The order of captions does not correspond to the images.

Figure 2, the order of captions does not correspond to the images. Figure 3, the order of captions does not correspond to the images.

This article is filled with too many errors, confusions, and omissions. The reviewing can be only continued after careful checking and improving by the author

1. The English of this paper needs to be greatly improved. Many sentences are not rigorous and standardized, which made them hard to be understood.

Author Response

Reviewer 1

This is an interesting work, which should be published in IJMS.

We thank the reviewer for acknowledging the quality of our work.

But there are some major comments, especially in writing statements.

  1. The English of this paper needs to be greatly improved. Many sentences are not rigorous and standardized, which made them hard to be understood. Line 26, 119,

As requested, we have edited the entire manuscript with a professional English native speaker. We have carefully proofread, check spelling, grammar, sentence structure and terminology.

  1. Only the results should be placed at the part of Results, no description of methods, Line 121-124, 337-339. No description of backgrounds, Line 204-205, 316-318.

As suggested by the reviewer, most of the sentences describing methods have been moved to the Materials and Methods section.

  1. Line 129-137? The writing formats of the whole paper should be cheeked carefully before submitting by the authors, especially the corresponding author!

As requested, we have edited the entire manuscript with a professional English native speaker. We have carefully proofread, check spelling, grammar, sentence structure and terminology.

  1. Figure 1, the title does not match the contents. The order of captions does not correspond to the images.

The figure 1 title and legend have been corrected

Figure 2, the order of captions does not correspond to the images. Figure 3, the order of captions does not correspond to the images.

Figure 2 and 3 legends have been corrected.

This article is filled with too many errors, confusions, and omissions. The reviewing can be only continued after careful checking and improving by the author

Comments on the Quality of English Language

  1. The English of this paper needs to be greatly improved. Many sentences are not rigorous and standardized, which made them hard to be understood.

As requested, we have edited the entire manuscript with a professional English native speaker. We have carefully proofread, check spelling, grammar, sentence structure and terminology.

Reviewer 2 Report

1. L34: Keywords should be key terms that did not show in the manuscript title, so "CRISPR/Cas9" and "melon" are not necessary.

2. L40: "n" an "x" should be italics.

3. Avoid using first-person writing throughout the manuscript.

4. Figure 1: The layout needs to be fixed, particularly Figure 1L, but the numbering of panels has a problem. It should be "I" following "H". Significant differences between means are missing in Figures 1F and 1L.

5. Figures 2C and 2E: The text should be reorganized for their layout and subscript.

6. L220: What do you mean by "AVGËŸAgNO3"?

7. Figure 3: Significant differences between means are missing.

8. Tables 1, 4, and 5: Significant differences between means are missing.

9. Statistical analysis: How many replicates were performed in each experiment?

10. It needs a conclusion section.

Author Response

L34: Keywords should be key terms that did not show in the manuscript title, so "CRISPR/Cas9" and "melon" are not necessary.

As requested, the keywords have been removed from the list.

  1. L40: "n" an "x" should be italics.

The mistakes have been corrected.

  1. Avoid using first-person writing throughout the manuscript.

As suggested, we have edited the entire manuscript with a professional English native speaker. We have carefully proofread, check spelling, grammar, sentence structure and terminology.

  1. Figure 1: The layout needs to be fixed, particularly Figure 1L, but the numbering of panels has a problem. It should be "I" following "H". Significant differences between means are missing in Figures 1F and 1L.

As requested, the numbering of the panels has been modified and the statistical test has been added.

  1. Figures 2C and 2E: The text should be reorganized for their layout and subscript.

As requested by the reviewer, the text has been reorganized.

  1. L220: What do you mean by "AVGËŸAgNO3"?

By “AVG*AgNO3” we mean a treatment with both AVG and AgNO3. For more clarity, we know wrote “AVG+AgNO3”.

  1. Figure 3: Significant differences between means are missing.

As requested, the statistical test was added.

  1. Tables 1, 4, and 5: Significant differences between means are missing.

The cited tables report transformation and base editing efficiency in different melon accessions. We are not comparing the transformation and base editing efficiency between the melon accessions and thus we did not perform statistical test. Because we have only few transgenic melons, we think that reporting transformation and base editing efficiency in different melon accessions is more suitable to our data.

  1. Statistical analysis: How many replicates were performed in each experiment?

As requested, we added the statistical tests and number of replicates in figures 1-3.

  1. It needs a conclusion section.

As requested by the reviewer, we added a conclusion section after the Material and methods section.

Round 2

Reviewer 1 Report

Thank the authors about their effort to improve the manuscript. The authors have comprehensively revised this paper. The English writing statements have improved significantly. It should be published after some minor revisions.

1. All figures, please carefully check that the order of captions does not match the order of images.

E.g. Figure 1, “E, F. The percentage of explants expressing DsRed at different inoculation times (E) and concentra-192 tions of Agrobacterium (F). G. Melon explants and Agrobacterium suspension in the 20 mL syringe. H. 193 Vacuum infiltration of Agrobacterium into melon cotyledon explants.”

The Figure annotation of “E, F” should be the image of “G, H”, but not image “E, F”.

2. The number of edited bases from the start codon should be gave out as a subscript followed the base letters. “editions (C-to-T and C-to-G) in the CmeIF4E gene.”

3. Table 2, the cultivar names and “Transformed/non-transformed” should be indicated in the tables.

4. Table 3 gives only the information of Primer sequences, but the title is “Segregation analysis of transgene expression in the T1 progeny of the transformant lines.” Match?

5. Figure 6, Each number or symbol in the figure should have a corresponding explanation. WT, 1-1, 3-6 indicates what? WT, T0, T1 indicates what? How to distinguish between two “T1” in Figure 6D? Some more detailed annotation is needed in the figure caption.

6. The expression of Reporter gene can represent the transformation efficiency? Why do not compare the number or proportion of transformed plants?

7. An important aspect of this article is about transformation efficiency.

“Our results showed that immersing the explants in the inoculation medium for 20 minutes significantly enhanced transformation efficiency. During the co-culture step, the use of filer paper, 10 mM 2-(N-morpholino)-ethanesulfonic acid 26 (MES)), and a temperature of 24°C significantly enhanced the melon transformation efficiency.”

Only 20min, 40min, and 60min were tested, how about the significance?

Only 24°C and 27°C were tested, how about the significance?

8. The conclusion is too long to get the key information. A summation including the best transformation condition bansed on this study and its successful application in CRISPR/Cas9-mediated base editing in Melon is enough.

Author Response

Thank the authors about their effort to improve the manuscript. The authors have comprehensively revised this paper. The English writing statements have improved significantly. It should be published after some minor revisions.

We thank the reviewer for all his comment and the time spent to carefully review our work.

  1. All figures, please carefully check that the order of captions does not match the order of images.

E.g. Figure 1, “E, F. The percentage of explants expressing DsRed at different inoculation times (E) and concentrations of Agrobacterium (F). G. Melon explants and Agrobacterium suspension in the 20 mL syringe. H. Vacuum infiltration of Agrobacterium into melon cotyledon explants.”

The Figure annotation of “E, F” should be the image of “G, H”, but not image “E, F”.

Thank you for pointing this mislabeling. The figure legend and the main text have been corrected.

  1. The number of edited bases from the start codon should be gave out as a subscript followed the base letters. “editions (C-to-T and C-to-G) in the CmeIF4E gene.”

As requested, the number of edited bases was edited as follow: CmeIF4EC322T/C323T and CmeIF4EC322T/C323G.

  1. Table 2, the cultivar names and “Transformed/non-transformed” should be indicated in the tables.

As requested by the reviewer, the table 2 was modified.

  1. Table 3 gives only the information of Primer sequences, but the title is “Segregation analysis of transgene expression in the T1 progeny of the transformant lines.” Match?

Thank you for pointing this mislabeling. The title of table 3 was modify.

  1. Figure 6, Each number or symbol in the figure should have a corresponding explanation. WT, 1-1, 3-6 indicates what? WT, T0, T1 indicates what? How to distinguish between two “T1” in Figure 6D? Some more detailed annotation is needed in the figure caption.

As requested by the reviewer and for more clarity, the figure 6 and its caption were modified.

  1. The expression of Reporter gene can represent the transformation efficiency? Why do not compare the number or proportion of transformed plants?

We have assessed the suitability of utilizing the red fluorescent protein DsRed as a visual indicator for genetic transformation of melon. Transient and stable expression of DsRed in melon explants was readily detected by fluorescence microscopy, allowing easy confirmation of gene introduction. DsRed's red fluorescence was distinctly visible under fluorescence microscopy, with no interference from background fluorescence. Additionally, the seeds and fruits that exhibited DsRed expression could be readily distinguished even under normal white light due to the distinctive color confered by the transgenic product. These findings suggest that DsRed fluorescence is a suitable proxy for evaluating the effectiveness of genetic transformation of melon plants.

  1. An important aspect of this article is about transformation efficiency.

“Our results showed that immersing the explants in the inoculation medium for 20 minutes significantly enhanced transformation efficiency. During the co-culture step, the use of filer paper, 10 mM 2-(N-morpholino)-ethanesulfonic acid 26 (MES)), and a temperature of 24°C significantly enhanced the melon transformation efficiency.”

Only 20min, 40min, and 60min were tested, how about the significance?

Prolonging the duration of Agrobacterium inoculation in genetic transformation has been proven to enhance the efficiency of transformation in various plant species. Nonetheless, the application of extended inoculation times is restricted due to the activation of defense mechanisms in plant tissues, leading to necrosis or programmed cell death. As a result, the regeneration of explants, and subsequently the efficiency of transformation, is negatively impacted. Furthermore, a prolongated inoculation time induces more Agrobacterium growth on the explant and the need of a higher antibiotic concentration to eliminate the Agrobacterium. In our case, we tested 3 inoculation times and found that 20 min of Agrobacterium inoculation yields the highest transformation efficiency.

Only 24°C and 27°C were tested, how about the significance?

In 2022,  some papers reported melon genetic transformation by CRISPR/Cas9 (Giordano et al., 2022; Liu et al., 2022; Leibman et al., 2022; Pechar et al., 2022). These genetic transformation protocols reported an inoculation temperature between 24 to 28°C. In our experiments, we choose to evaluate 24°C and 27°C and found that a temperature of 24°C gives better melon transformation efficiency.

We acknowledged the reviewer's suggestion to test a wider range of temperatures. However, expanding the range of tested parameters would significantly increase the experimental time and manpower required. We thank the reviewer for his understanding.

  1. The conclusion is too long to get the key information. A summation including the best transformation condition bansed on this study and its successful application in CRISPR/Cas9-mediated base editing in Melon is enough.

As requested by the reviewer, we have reduced the length of the conclusion and focused on the key informations.

Reviewer 2 Report

It has been improved and I don't have further questions.

Author Response

We thank the reviewer for the careful reading of our revised text.